# The representational hierarchy in human and artificial visual systems in the presence of object-scene regularities

**Stefania Bracci**[1,2]*, **Jakob Mraz**[2], **Astrid Zeman**[2,3], **Gaëlle Leys**[2], **Hans Op de Beeck**[2]

**1** Center for Mind/Brain Sciences—CIMeC, University of Trento, Rovereto, Italy, **2** Brain & Cognition Research Unit, Leuven Brain Institute, KU Leuven, Leuven, Belgium, **3** Mathematics and Statistics, University of Melbourne, Carlton, Australia

* stefania.bracci@unitn.it

**Data Availability Statement:** Due to ethics restrictions (ethics protocol B322201630276), raw brain images cannot be made available. All pre-processed brain functional images (beta images),

## Abstract

Human vision is still largely unexplained. Computer vision made impressive progress on this front, but it is still unclear to which extent artificial neural networks approximate human object vision at the behavioral and neural levels. Here, we investigated whether machine object vision mimics the representational hierarchy of human object vision with an experimental design that allows testing within-domain representations for animals and scenes, as well as across-domain representations reflecting their real-world contextual regularities such as animal-scene pairs that often co-occur in the visual environment. We found that DCNNs trained in object recognition acquire representations, in their late processing stage, that closely capture human conceptual judgements about the co-occurrence of animals and their typical scenes. Likewise, the DCNNs representational hierarchy shows surprising similarities with the representational transformations emerging in domain-specific ventrotemporal areas up to domain-general frontoparietal areas. Despite these remarkable similarities, the underlying information processing differs. The ability of neural networks to learn a human-like high-level conceptual representation of object-scene co-occurrence depends upon the amount of object-scene co-occurrence present in the image set thus highlighting the fundamental role of training history. Further, although mid/high-level DCNN layers represent the category division for animals and scenes as observed in VTC, its information content shows reduced domain-specific representational richness. To conclude, by testing within- and between-domain selectivity while manipulating contextual regularities we reveal unknown similarities and differences in the information processing strategies employed by human and artificial visual systems.

## Author summary

Computational object vision represents the new frontier of brain models, but do current artificial visual systems known as deep convolutional neural networks (DCNNs) represent the world as humans do? Our results reveal that DCNNs are able to capture important representational aspects of human vision both at the behavioral and neural levels. At the

anatomical masks (ROIs), statistics data, and the Matlab code, will be made available through the Open Science Framework at https://osf.io/u59ez/.

**Funding:** S.B. was funded by FWO (Fonds Wetenschappelijk Onderzoek) through a postdoctoral fellowship (12S1317N) and a Research Grant (1505518N). J.M. was supported by the Ad futura Scholarship of the Public Scholarship, Development, Disability and Maintenance Fund of the Republic of Slovenia. H.O. B. was supported by the KU Leuven Research Council (C14/21/047; AKUL/19/05) and FWO grants EOS HumVisCat (nr. 30991544) and G0D3322N.

**Competing interests:** The authors have declared that no competing interests exist.

behavioral level, DCNNs are able to pick up contextual regularities of objects and scenes thus mimicking human high-level semantic knowledge such as learning that a polar bear "lives" in ice landscapes. At the neural representational level, DCNNs capture the representational hierarchy observed in the visual cortex all the way up to frontoparietal areas. Despite these remarkable correspondences, the information processing strategies implemented differ. In order to aim for future DCNNs to perceive the world as humans do, we suggest the need to consider aspects of training and tasks that more closely match the wide computational role of human object vision over and above object recognition.

## Introduction

We live in a structured world; as a consequence, sensory input is not a random collection of lines and patterns but can be organized into meaningful and identifiable wholes such as objects. These objects show particular relationships to their environment. Some objects are most likely to appear in specific contexts, such as penguins in ice landscapes or lions in the savannah. This set of rules, similarly to the grammar of our language, provides a structure to guide our behaviour [1]. Here we investigate how information about object domain and their cross-domain structure is represented through the visual processing in human and artificial neural systems.

While visual information is fully intertwined at the retina level, soon through the human hierarchical visual pathway, information pertaining to the different object domains is processed in largely separated brain networks [e.g., 2,3] as confirmed by human neuroimaging revealing rich domain-specific object spaces characterizing the different processing channels. As an example, behaviourally relevant image dimensions are encoded in object and scene areas: animate features such as the eyes/mouth, useful for identifying living entities in the former [4–6], and spatial layout, informative for scene navigation in the latter [7–9]. At the same time, statistical regularities of the world influence perception [10–13] as shown by faciliatory effects of context observed during object recognition [14] and evidence for interaction between object and scene processing at the neural level [15–18]. While such studies indicate that the representation of objects might be influenced by whether they share similar context statistics, to our knowledge, none of them included a multivariate test pinpointing the representational similarity of neural patterns across objects and scenes. Is the frequent occurrence of a penguin in ice landscapes sufficient to modulate how penguins and ice landscapes are coded despite the separate processing channels? Or does interaction occur at a later stage of visual processing when information becomes relevant to support goal-directed behaviour?

Even less is known about how this across-domain segregation and integration happens in artificial vision models. In the past decade, artificial computer vision models have been developed that are able to classify visual patterns with human-level performance, such as deep convolutional neural networks [DCNNs; 19] and recent adaptations including residual, recurrent, and transformer networks [20]. These artificial models appear to develop representations similar to those in human visual cortex [21,22]. Various avenues have been suggested to further improve this correspondence, such as the use of recurrent processing and changes in training regimes [20,23]. But does the representational hierarchy learnt by DCNNs mimic the information transformation emerging through the human visual hierarchy? The presence of object-scene regularities potentially provides a critical test of this correspondence. Many DCNNs employed in computational vision are trained on object recognition which is considered the main computational goal of visual cortex [24] and employ large image sets in which each

image is attached to a single label [25]. In such scenario, DCNNs trained in object recognition do not need to explicitly separate objects from background scene; short cuts are available and sufficient to solve the problem. In other words, visual information from image background might be equally useful to succeed at the task at hand, thus shaping the learnt object space. This suggests that DCNNs might very well mimic the ability of the human perceptual system to take advantage of image statistical and contextual regularities experienced during lifetime [11,14,26], but these artificial models might find this solution in a different way without explicitly segregating object and scene information.

Here, we compare "neural" representations in the human brain and in artificial DCNNs using an original stimulus set that includes object images and background images with a variety of object/scene domain specific properties, as well as a manipulation of object-background co-occurrence in real-world images. The results show that DCNNs in a quite remarkable way, appear capable of mimicking conceptual-like human knowledge of the world such as capturing the conceptual similarity for a specific object-scene pair (gorilla and jungle forest) as well as the hierarchical representations observed along the human visual pathway, all the way up to frontoparietal areas. At the same time, results suggest difference in the underlying computational strategy implemented by the two systems.

## Results

To test how the biological or artificial brain represent object within-domain and cross-domain regularities through their computational hierarchy, we created a stimulus set which includes two category domains of contextually related pairs of images (Fig 1). Each pair includes one animal (clownfish, ladybug, passerine bird, seagull, polar bear, and gorilla) and its associated background scene (anemone, leaves, tree branches, seashores, ice landscapes, and jungle forests). We carefully selected images of animals that have a neutral background (e.g., the polar bear and the gorilla have a similar grayish background) to avoid possible confounds where the object background might be informative for recognition.

This unique stimulus set allows us to differentiate between alternative hypotheses about how objects and scenes might be represented (Fig 2). The *domain model* predicts a separation between animals and scenes. On the contrary, the *co-occurrence model* predicts contextually related associations between each animal and its specific background scene. These two models are orthogonal to each other (r = -0.05). Two additional control models rule out the role of visual factors: (1) The *GIST model* [27], a good descriptor of scene statistics [13], well describes responses in lower visual areas [28]; (2) the *condition model* captures within-category similarities for each animal and scene condition. Representational similarity analysis [RSA; 29] allows to test each model's ability to capture the object space in the brain and DCNNs.

### Object and scene information is processed in separate visual pathways, with contextual effects emerging in frontoparietal areas

How does the human brain encode both object-scene separation and interaction? To test this, we analyzed fMRI scans from participants (N = 19; two long scan sessions per participant) while they watched the stimuli from Fig 1A in an event-related design. Participants were asked to indicate to what extent each image would normally co-occur with the previous image. This task allows us to investigate representations in both task-independent and task-dependent areas [30,31]. We investigated the representational structure along the hierarchy of visual regions from the primary visual cortex over regions involved in object and scene perception up to regions in the frontoparietal cortex that encode flexible goal-directed representations [32] and that are known to be connected to domain-specific regions [33].

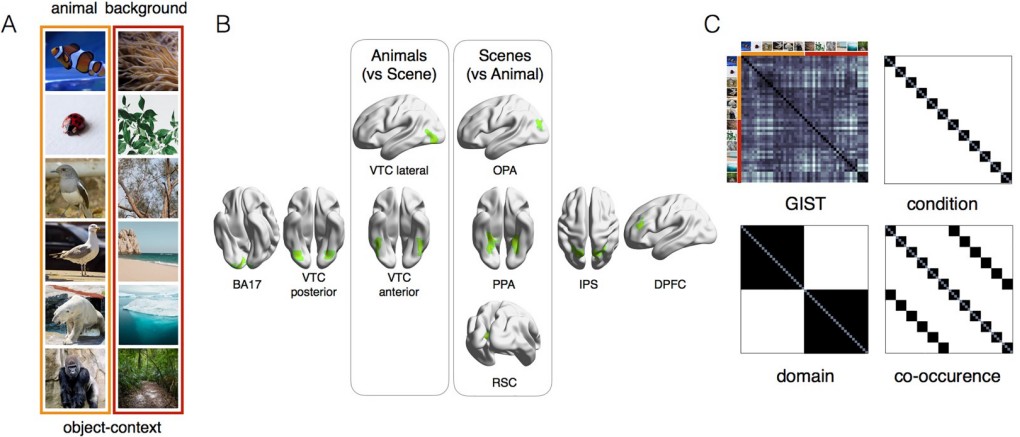

**Fig 1. Experimental design.** (A) The stimulus set includes 2 domains: animals and scenes, each including 6 different identity conditions (4 images for each condition). Due to copyright restrictions, the images shown here are royalty-free examples of images downloaded from https://unsplash.com/ chosen based on same criteria used to select the original stimuli. The pictures of animals were carefully selected to avoid that background information could be informative for object identification (e.g., the polar bear and the gorilla have a very similar neutral background). To control for shape, we further divided the animal categories in three subsets along the animacy continuum (2 mammals, 2 birds, and 2 small rounded animals). Within each subset, animals are matched for body shape (e.g., gorilla and polar bear) but each animal is paired with a different scene. As an example, the passerine bird and the seagull have similar body shape but are associated with two different backgrounds. As for the pictures of scenes, 3 of the backgrounds are characterized by rich navigational properties where there is no object in focus in the middle of the image: seashores, ice landscapes, and jungle forests. The other 3 backgrounds are object-like scenes with little navigational layout properties: anemones, leaves, and tree branches. Concurrently, animals and scenes conditions were selected based on their frequent co-occurrence in real-world images: polar bears live in ice landscapes and gorilla live in forest jungles, thus allowing the creation of 6 specific object-scene contextual pairs (B) The ROIs included in the brain RSA analysis included visual areas (for their relevance in object recognition) and frontoparietal areas (for their relevance in goal-directed behavior): BA17, posterior ventral-temporal cortex (VTC), anterior VTC, lateral VTC, occipital place area (OPA), parahippocampal area (PPA), retrosplenial cortex (RSC), intraparietal sulcus (IPS), and dorsal prefrontal cortex (DPFC). See Methods for details on the localization procedure. (C) Four models were tested: GIST, condition, domain, and co-occurrence.

Results in visual cortex reveal a clear separation between animal and scene representations, despite their co-occurrence in our visual experience. The domain model captures most variance in many areas from posterior to anterior temporal cortex but, importantly, not in early visual cortex BA17 where only the GIST model reached significance (z = 0.14; p = 0.0001, Fig 2A). The separation between animals and scenes is particularly evident in animal-selective areas (domain model: z > 0.84; for all ROIs, p < 0.0001, Fig 2A) with an additional minor contribution of the GIST model in posterior VTC (z = 0.17; p < 0.0001, Fig 2A). This separation is also very obvious when visually inspecting the dissimilarity matrices displayed in Fig 2 (bottom). In scene-selective areas, the domain model also best captures the representational structure in 2 out of 3 ROIs (PPA: z = 0.46; OPA: z = 0.29; both p < 0.0001, Fig 2A). The remaining models did not explain any additional variance in these areas. The very same results were observed using single-subject data (Fig 2B).

The information comes together again when moving downstream, with a small but significant effect of the co-occurrence model in frontoparietal areas (IPS: z = 0.18; DPFC: z = 0.16, both p < 0.0001, Fig 2A), thus shifting the representational content in these ROIs from object-scene separation to object-scene contextual association. When using individual subject data, the effect for the co-occurrence model in these ROIs is relatively small but highly significant

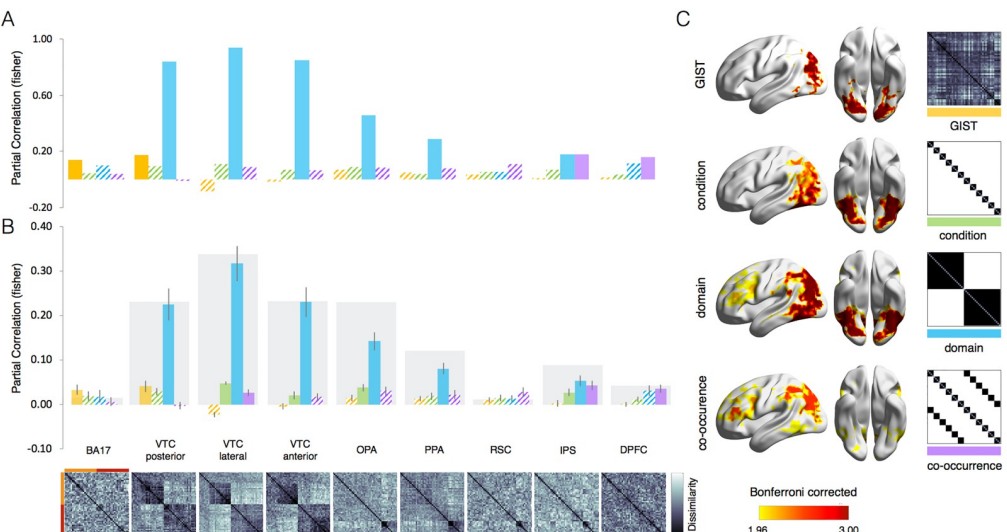

**Fig 2. The representational hierarchy for separation and interaction of objects and scenes in the brain.** The ROI-based (A, B) and whole-brain (C) RSA results for the 4 models (GIST, condition, domain, co-occurrence) are shown for brain data. Results reveal a strong separation for domain (scene and animal) representations in most ventral regions. The effect for animal-scene co-occurrence emerges in frontoparietal areas. (A) For group-averaged results, filled bars indicate significant values against baseline (p < 0.001) computed with permutation tests (10,000 randomizations of stimulus labels). (B) For individual subject results, reliability boundaries (in gray) indicate the highest expected correlation considering signal noise (see Methods) and error bars indicate SEM. Filled bars indicate significant values against baseline (p <0.005, corrected for n. or ROIs) calculated with pairwise t-tests across subjects (n = 19). For each ROI, the neural dissimilarity matrix (1—r) is shown below. (C) The random-effects whole-brain RSA results corrected with Threshold-Free Cluster Enhancement [TFCE; 37] are displayed separately for each individual model against baseline [BrainNet Viewer; 38]. Note that for some of these maps (e.g., co-occurrence vs domain), the direct contrast did not reveal a significant difference.

(DPFC and IPS: z > 0.04, both p < 0.005, Fig 2B). It also reaches reliability boundaries (IPS: 0.09; DPFC: 0.04), which indicates the highest expected correlation in a brain region after accounting for signal noise [34]. In DPFC, the remaining models did not explain any significant variance in the patterns, while in IPS there were significant correlations for the domain model (z = 0.18; p < 0.0001, Fig 2A; z = 0.05; p < 0.0001, Fig 2B) and condition model (z = 0.03; p = 0.001, Fig 2B). Together, these results show that the overall representational content in visual cortex largely distinguishes category information at domain level (object vs scene), with interaction of these components emerging at a later processing stage in regions known to support goal-directed behavior [32]. The whole-brain RSA confirmed these results revealing that the domain model strongly activates regions in the ventral and dorsal visual pathways, whereas the co-occurrence effect for animals and matching scenes is confined within frontoparietal areas (Fig 2C). We note that, although the distribution of strongest effects in Fig 2C suggests a shift in emphasis towards frontoparietal cortex for the co-occurrence relative to the other models, this shift did not result in significant differences in a whole-brain analysis. Finally, to evaluate the potential impact of metrics' choice, we re-run the RSA with two alternative distance measures: the cross-validated Mahalanobis distance (following [35]) and the Euclidean distance. Results confirmed the RSA analysis performed with 1 minus correlation (S1 Fig). In line with results reported in [36], these three metrics tend to give similar results, and discrepancies are only noted in very specific designs.

## The representational hierarchy in DCNNs mimics representational transformations observed in the human brain

DCNNs widely employed in image classification [39], are now considered a promising venue to explain biological object vision at the neural level, proving to match human object recognition abilities in several benchmarks [40–42]. Yet, it is unclear the extent to which machine vision learns brain-like representational hierarchy and computational strategies [43]. To test this, we use RSA to evaluate the predictive power of our models in four DCNNs: Alexnet, VGG16, GoogLeNet, and ResNet-50. All models are trained in object recognition (ImageNet) which is widely considered the main computational goal of ventral visual cortex [24] and are the most frequent benchmarks used in the literature that suggests similarities in representational structure between human and computational vision [44,45].

Quite remarkably, results revealed high similarity between biological and artificial systems in their representational hierarchy (Fig 3A). At an early stage of computational processing, the

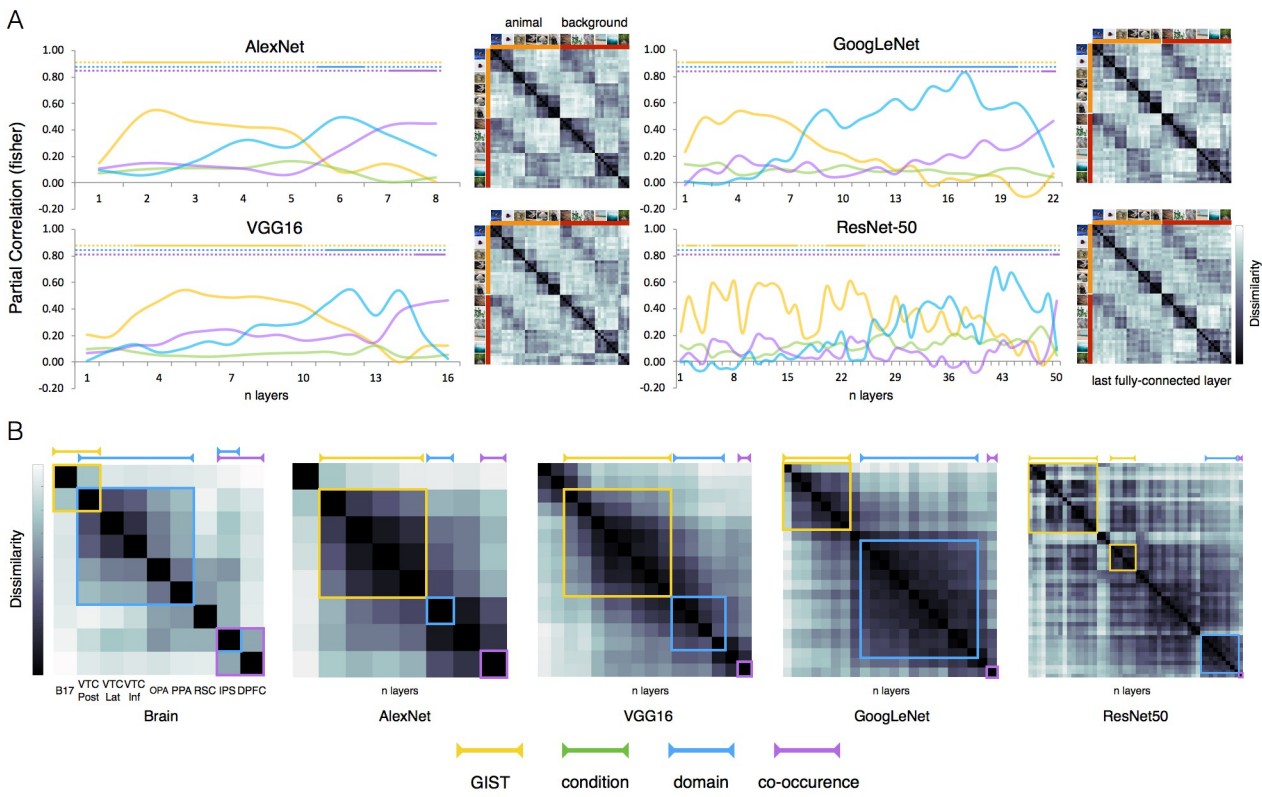

**Fig 3. Similar representational hierarchy in the brain and DCNNs.** (A) The DCNNs RSA results for the 4 models (GIST, condition, domain, co-occurrence) are shown for 4 DCNNs (AlexNet, VGG16, GoogLeNet, ResNet-50). The network's depth is shown on the x axis. For each graph and each model, color-coded lines indicate significant effects relative to all remaining models ($p < 0.001$) computed with pairwise permutations tests (10000 randomizations of stimulus labels). For each DCNN, the representational dissimilarity (1—r) is shown for the last fully connected layer. (B) Correlational matrices show second-order relationships among representational patterns in the brain's ROIs and individual DCNNs' layers. Color-coded line boxes highlight the ROIs and DCNNs' layers where each model reaches significance. For brain areas, significance for each model is shown relative to baseline ($p < 0.0001$), calculated with permutation tests (10,000 randomizations of stimulus labels). The order in which ROIs are shown does not imply a strict correspondence with the computational hierarchy in the brain. For DCNNs' layers, significance for each model is shown relative to all remaining models ($p < 0.001$), calculated with permutation tests (10000 randomizations of stimulus labels). Both systems show similar transformations in the representational space. Early on, the object space reflects image low-level visual properties (GIST model, yellow color-coded), it then shifts towards animal-scene domain division (domain model, light-blue color-coded), to finally reveal animal-scene co-occurrence effects (co-occurrence model, purple color-coded).

GIST model best predicts representations in all neural networks (p < 0.001 relative to all models). At mid/high-level layers, when the GIST model drops in performance, the domain model increases and reaches significance relative to the remaining models (p < 0.001). Finally, the representational space at the latest processing stage shows a salient object-scene association. For all networks, the co-occurrence model peaks at the final processing stages (i.e., fully connected layers) and, relative to the remaining models, explains significantly better the DCNNs' representational space (p < 0.001 relative to all models; for summary statistics, see Fig 3B). This structure is also visible in the secondary diagonal emerging in the lower left and upper right quadrant of the networks' representational dissimilarity matrices (Fig 3A). The entangled animal-scene representation emerges later through the network's hierarchy, ruling out the contribution of low-level image properties best captured by low-level visual models such as the GIST (e.g., visual similarities between each animal category and its associated environment).

Overall, these results suggest that DCNNs are able to capture the way visual information is transformed through the human brain visual hierarchy, from early visual cortex encoding low-level visual properties, through VTC encoding domain specific information in separate channels, all the way up to frontoparietal areas where information from the different domains is combined to support goal-directed behavior focused upon object-scene regularities (Fig 3B). These results were also confirmed in an follow up exploratory analysis where we directly correlated the hierarchical representational space for each DCNNs with the representational space in the selected ROIs as well as across the whole brain (S2 Fig).

## DCNNs capture humans' conceptual knowledge of object-scene co-occurrence

An object code that incorporates statistical regularities of an object in its background might be relevant to mimic human-like object recognition behavior, which indeed is influenced by the object-scene interaction [10,11]; context facilitates object recognition and vice versa [14]. Given that DCNNs representations at their final layers appear to capture a degree of contextual regularities between an object and its recurring background scene, we might expect similarities with human behavior when humans judge image similarities in terms of co-occurrence (see Methods).

Results confirm this prediction. The correlation between human co-occurrence judgments and DCNNs' object space (Fig 4A) increases throughout the DCNNs' processing hierarchy to reach its peak at the final processing stages for all DCNNs (AlexNet: z = 0.46; VGG16: z = 0.59; GoogLeNet: z = 0.65; ResNet-50: z = 0.56). The DCNNs' ability to capture human behavior is also visible when inspecting the MDS's space (Fig 4B). In a similar fashion as the object space generated by behavioral judgments, the DCNNs object space in their final layer shows an orderly structure: each animal (e.g., yellow circle) sits close to its semantically associated scene (e.g., yellow square), thus revealing a clear contextual effect. To confirm this observation statistically, we tested similarities for congruent pairs (polar bear and iceberg) versus the average of all remaining incongruent pairs (polar bear and jungle). We observed significantly higher similarity for congruent relative to incongruent pairs in behavioral (p < 0.0001; Fig 4C) as well as DCNNs data (for all models, p < 0.001). Thus, DCNNs' sensitivity to image statistical regularities of objects occurring in typical environments results in the acquisition of a "representation of the world" that closely resembles humans' conceptual judgments.

## Knowledge about object-scene co-occurrence is reached through different computational strategies in DCNNs

The ability of DCNNs to capture human high-level conceptual knowledge about the world is striking, but does this prove that the artificial networks have a real understanding of objects

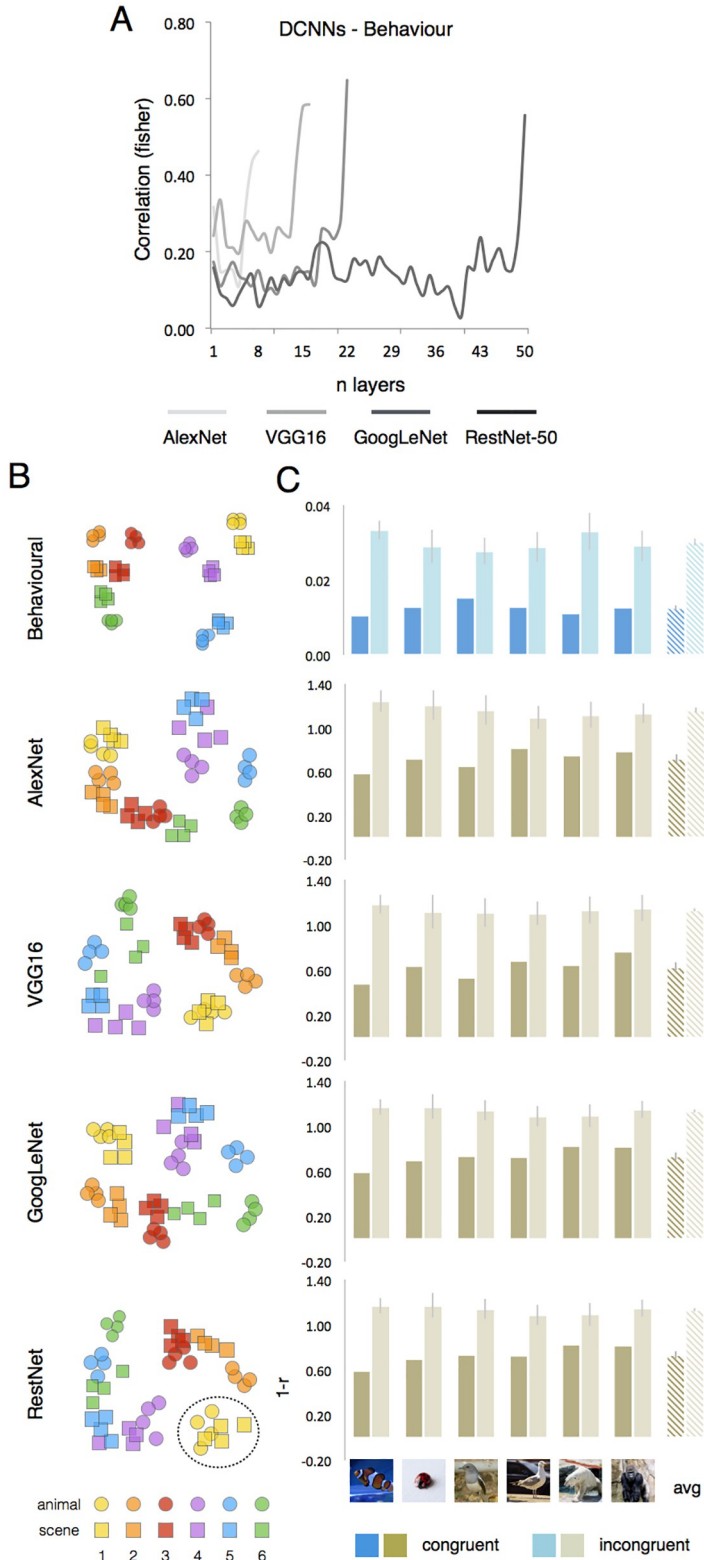

**Fig 4. DCNNs acquire human-like conceptual biases.** (A) The correlational plot shows the degree of similarity between behavioral judgments and each of the four DCNN architectures. (B) MDS spaces (metric stress) show

consistent object-scene clusters in human behavioral judgments as well as DCNNs (last fully connected layer). (C) The object-scene cluster analysis (right) for behavioral and DCNNs (last layer) data, show a consistent significant effect of congruency (lower distance) for object-scene stimulus pairs. Dashed bars show the averaged data for the 6 stimulus pairs.

and scenes? One possibility is that object-background segregation might be less optimized in DCNNs architectures trained with natural images (e.g., ImageNet) where regularities between objects and backgrounds are common. For instance, a whale lives in open water, not on the street. As such, any image feature (e.g., not necessarily object-specific) might become a useful piece of information to recognize the object [46]. Here, we test this hypothesis by systematically manipulating object/background co-occurrence during DCNNs training (see Methods): from 0% co-occurrence (objects within the same category were no more likely to occur on the same background type than objects from different categories) to 100% co-occurrence (objects within the same category were always presented on the same background). Confirming the role of image regularities in facilitating object recognition (Fig 5A), the model's validation test revealed significantly higher model performance for the 100% co-occurrence condition (accuracy: 0.85) relative to remaining conditions (accuracy < 0.72; t(4) 3.95, p < 0.02, for all pairwise t-tests). Furthermore, we predicted an increasing representational bias towards humanlike conceptual object-scene associations in accordance with increasing object-scene co-occurrence. We run an RSA analysis with each of our predicting models and tested correlations for fully connected layer 7 in a 4x4 ANOVA within Co-occurrence (0%, 58%, 83%, 100%) and Model (GIST, condition, domain, co-occurrence) as within-subject factors. Results revealed a significant Co-occurrence x Model interaction (F(9,36) 11.82, p < 0.0001; Fig 5B) showing that different levels of object-background regularities in the training set result in differences in DCNN's representational content (fc7) as captured by our models. These results are important to also confirm the effects observed in DCNNs (Fig 3) through a much larger stimulus set.

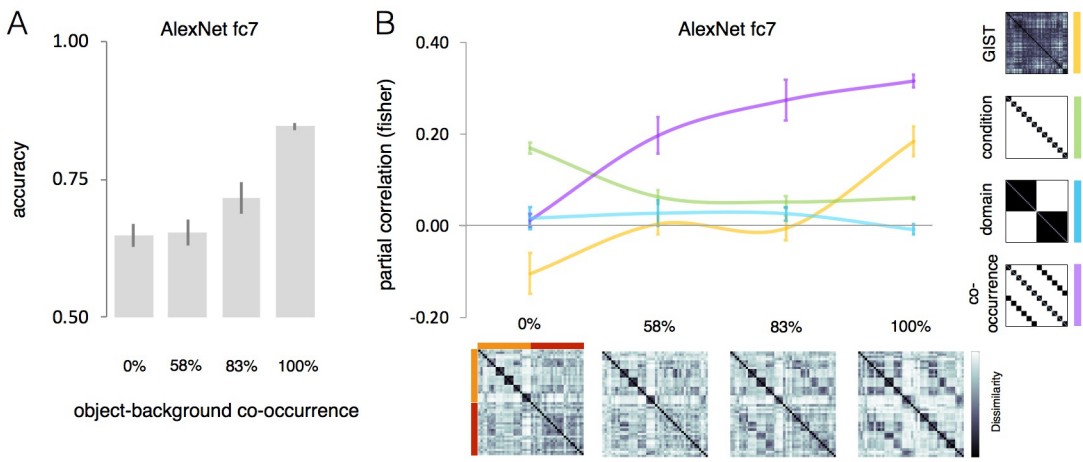

**Fig 5. The effect of increasing levels of object-scene co-occurrence in DCNN object space.** We trained a DCNN multiple times (N = 5) in 4 training conditions with increasing levels of object-background regularity from 0% to 100%. (A) Accuracy results for the model's validation test. (B) The RSA results for the 4 models (GIST, condition, domain, co-occurrence) are shown for AlexNet's fc7 that underwent different training regimes with increasing co-occurrence (0%, 58%, 87%, 100%) between objects and backgrounds (n = 6). Error bars indicate SEM. In the dissimilarity matrices, orange represents animal conditions and red represents background conditions.

Follow-up analyses revealed that the contextual-related effect is absent when the training is based on random object-scene associations (0% co-occurrence: $z = 0.01$) but emerges when regularities in the training set increase (58%: $z = 0.20$; 83%: $z = 0.27$; 100%: $z = 0.32$). This effect is already significantly higher for the 58% condition relative to 0% condition (t(4) 3.68, $p = 0.02$) and it increases for higher levels of object-scene co-occurrence (83%: t(4) 6.51, $p = 0.003$; 100%: t(4) 20.22, $p < 0.0001$; both relative to 0%). Interestingly, the effect of the condition model is significantly high when the network cannot rely on scene information (0%: $z = 0.17$), but strongly decreases when regularity increases (58%-100%: $z < 0.06$; t(4), $> 6.24$, $p < 0.004$, for all 3 comparisons). In the 100% co-occurrence condition, the correlation with the GIST model reaches significance ($z = 0.18$; t(4), $> 5.78$, $p = 0.004$), probably because in this condition there is an increase of lower-level background features that can be relied upon. This last condition shows that finding a representation of object-scene correspondences can be obtained through various processing strategies. This motivated us to take closer look at the processing stages right before the stage at which object-scene correspondences are represented.

## Multiple object (domain-specific) spaces in the visual cortex but not in DCNNs

In visual cortex, the domain model captures most of the variance in animal as well as in scene areas, but the dissimilarity matrices (Fig 2A) reveal a marked difference between the two sets of regions that likely reflect differential domain-specific object spaces [7–9,47,48]. This rich dimensionality can support the need of our brain to employ different representations for different behavioural needs [49]. For instance, in scene selective areas, the degree of navigational layout well characterizes its representational content which is relevant for naviation [7,8]. In a similar fashion, in animal selective areas, the degree of animacy [47,48], and the animal-specific features [4,5] might be relevant to support social-related computations. In our results, DCNN's mid-layers show domain division for animals and scenes (Fig 3A), but does this division embed rich domain-specific object spaces like those observed in the human visual cortex? We tested two domain-specific dimensions that well characterize the object space in animal and scene areas: the animacy continuum for the animal domain and navigational layout information for the scene domain. These two dimensions were included in our stimulus set (Fig 6A) and are captured by the animacy continuum and the navigational layout model, respectively (see Methods). To test the ability of DNNs, trained on object recognition, to capture domain-specific object spaces in scene and animal selective areas, as above we used RSA to test the two domain-specific models (animacy continuum and the navigational layout) and two control models (condition and GIST) in brain and DCNNs data. To account for the different domains we tested two instances of the same DCNNs architecture (GoogLeNet), trained to classify the basic level category in the two domains: objects (ImageNet) and scenes (Scene 365).

As expected, the RSA on the brain data (Fig 6B, left) confirms differential representational spaces reflecting domain-specific object spaces in visual cortex. Whereas VTC areas show a significant preference for the animacy continuum model (VTC post: $z = 0.76$; VTC lat: $z = 0.84$; VTC inf: $z = 0.70$; all $p < 0.001$, relative to the remaining models), scene-selective PPA and OPA show a significant preference for the navigational layout model (OPA: $z = 0.47$; PPA: $z = 0.38$; both $p < 0.001$, relative to the remaining models). Results were replicated in a follow-up analysis, where each domain-specific space (upper quadrant RDM for animals and lower quadrant RDM for scenes) was tested separately (S3 Fig). Together, these results highlight the functional specialization of the human brain for multiple domain-specific

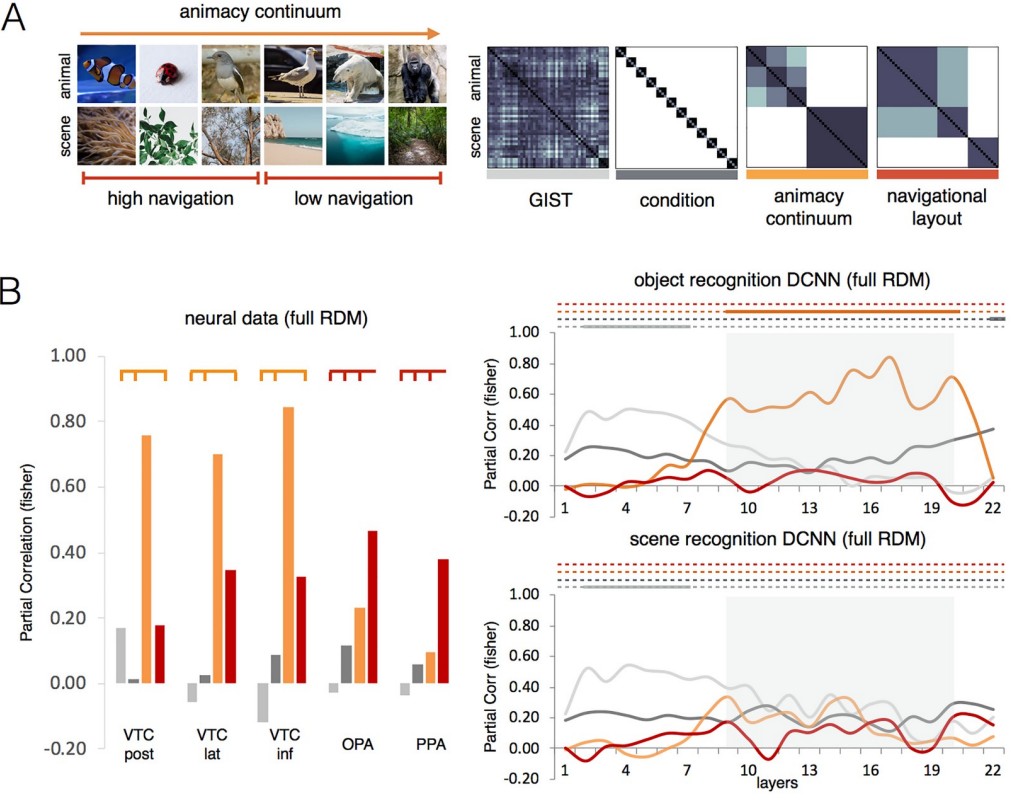

**Fig 6. Multiple domain-specific object spaces.** The RSA results for the 4 models (GIST, condition, animacy continuum, navigational layout) are shown for group-averaged brain (left) data and DCNNs (right). For the neural data, only ROIs where the domain model reached significance were included (see Fig 2A). The same DCNN architecture (GoogLeNet) was trained either on object recognition (ImageNet), or scene recognition (Scene 365). For comparison with the first RSA analysis (see Fig 3A), gray shaded areas indicate the network's layers in which the domain model significantly outperformed the remaining models. Color-coded lines on top of bar/graphs indicate the network's layers/ROIs where each model significantly outperformed the remaining models (p < 0.001) computed with pairwise permutations tests (10000 randomizations of stimulus labels).

computations and it highlights a representational richness that is exactly what one expects in a system with separate streams to process animal and scene information relevant to support domain-specific computations.

Do DCNNs trained on category (objects or scenes) recognition learn representations that mimic brain domain-specific object spaces? To address this question, we tested the same architecture (GoogLeNet) trained either on object or scene recognition. The RSA for DCNN data trained on object recognition (GoogLeNet trained on ImageNet) partially replicates results observed in VTC showing a significant effect for the animacy relative to all remaining models in those same layers that revealed a domain-division effect in the previous RSA analysis (Fig 6B, gray-shaded areas). Interestingly, at the latest processing stage the representation was best predicted by the condition model which reflects the representation expected when a network learn to distinguish the different conditions at the basic-level. Overall, this result reveals higher sensitivity for the dimension DCNNs were trained on (note that the Imagenet dataset contains a high % of animal images) and points to a potentially critical role that training tasks play in

developing domain-specific spaces in DCNNs. On the contrary, the DCNN trained in scene recognition (GoogLeNet trained on Scene 365), throughout its layers, did not reveal any similarity with the object space observed in scene-selective areas (Fig 6B). It did not learn a representational space that reflects the amount of navigational layout features present in the image. This result is not totally unexpected; scene recognition tasks require the elaboration of information that substantially differs from the information necessary to support scene navigation computations, thus questioning the role of purely recognition-based training tasks in capturing the richness of object space observed in visual cortex. We note that the methods employed here allow us exploring the dominant dimensions that diverge/converge across brain and DCNNs data. Alternative methods [e.g., 50] might be more sensitive to detect less preponderant dimensions yet present in the data. Finally, the different training task did not influence the representations learnt in the early layers. In early layers, both object and scene recognition DCNNs showed a representational space that was best captured by the GIST model relative to the remaining models (Fig 6B). These results were replicated when the representational space for animals (RDM upper quadrant) and scenes (RDM lower quandrant) was tested separatedly (S3 Fig).

## Discussion

We investigated how human and machine object vision represent within-domain and across-domain information of animals and scenes matched for contextual regularities and to what extent the latter mimics human object vision computations. We measured the representational similarity within domain (for animals and scenes) as well as across domains (animal-scene association), in a stimulus set that includes object-scene pairs that often co-occur in the visual environment. Results revealed that DCNNs employed in computational vision are able to capture conceptual representations humans have about object-scene correspondences. Not only DCNNs trained to perform object recognition are able to capture human judgments about contextual associations between and animal and its typical scene, but most importantly, they show a good hierarchical correspondence at the neural level. Nevertheless, our results do also show differences in terms of implemented computational strategies. In the visual cortex, object and scene information is processed in separated pathways, which reveal domain-specific representational contents for animal (animacy continuum) and scene (navigational layout) processing. Interaction between object and scene components was observed at a later processing stage in areas that contribute to goal-directed behavior. DCNNs' mid-layers showed a similar degree of object/scene separation, but its information content shows reduced domain-specificity even when DCNNs were trained on domain-specific recognition tasks (i.e., scene recognition). Further, the emergence of human-like high-level conceptual representation of object-scene co-occurrence in DCNNs depends upon the amount of object-scene co-occurrence present in the image-set thus highlighting the fundamental role of training history. In sum, despite the remarkable achievements shown by convolutional object recognition trained DCNNs, when aiming to mimic the rich and multiple representational spaces observed in the human brain, future brain models should extent their focus beyond recognition tasks.

The primate brain is, at least in part, characterized by separated modules for domain-specific processing [51]. These modules are relatively independent such that confined lesions disrupt computations for one domain (e.g., animate entities) leaving unaffected other domains [e.g., inanimate objects; 52]. In visual cortex, domain-specific areas encode object dimensions tailored to support specific computations [33,53,54]. As an example, view-invariant features represented in face- [54,55] and hand-selective regions [56,57] reflect domain-specific computations: the former to support identity recognition [58], the latter to support action

understanding [56]. In agreement, our results show that in addition to a large division between animal and scene representations, within each domain, representational content reflects the type of computations these networks support: animacy features in animal-selective areas [4,5,59] and layout navigational properties in scene-selective areas [7,60]. We can show this representational diversity because in our study we included separate behavioral-relevant dimensions for objects (i.e., animacy continuum) and background scenes (i.e., navigational properties), while this was typically not done in previous studies. Furthermore, by including objects and backgrounds that co-occur in the environment, we also demonstrate that at the level of domain-specific representations there is not yet a strong effect of such statistical regularities. This is not to say that there is no interaction between object and scene representations in human object vision, which in fact, has been reported in previous studies [12,15,61,62]. Representations of scenes and objects are modulated by various statistics, including the extent to which objects co-occur in the same scene [18,63]. However, in these studies the neural responses to objects and scene were tested separately, and never directly compared across domains (e.g., from objects to scenes). Thus, the nature of such an interaction is still debated and it might occur on top of domain modularity for objects and scenes [64,65]. Our study points in this direction, showing that, ultimately, all such interaction effects ride on top of a fundamental division of labor for animal versus scene properties, in those areas supporting flexible goal-directed representations [30,31].

In a quite remarkable fashion, DCNNs capture both human behavior as well as the hierarchical representations observed in the human brain. When it comes to behavior, DCNNs pick up typical object-background regularities reflecting conceptual knowledge we have about the world, thus reaching the human-unique conceptual knowledge level. However, the information processing strategy might be very different in DCNNs. Indeed, when we manipulated object-background systematically in a smaller-scale training regime, even arbitrary object-background associations resulted in strong effects of such associations upon the representations that these networks develop (Fig 5). This result highlights the critical role played by training on the resulting object space. In the context of a one-image, one-label approach, any pixel in the image might contain features useful to recognize objects. When such regularities are prevalent in a dataset, as it is for natural image databases, shortcuts can be taken [66]. In these situations, background information becomes as relevant (or even more relevant) as the object to be recognized [67]. This points to a substantial difference between human vision and feedforward DCNNs. Although, faciliatory effects on object recognition have been observed when an object and its background are congruent [14], thus highlighting the potential role of statistical regularities in the environment in supporting human behavior [26], the ability to separate foreground objects from background information is a prerequisite to object recognition, for which the VTC plays a critical role. On the contrary, the object recognition DCNNs tested here, which are widely used in computational vision, in their final layers, do integrate background information in their learnt object space.

The effect of background on the object space in DCNN's fully connected layers raises the question of how much of the previously reported domain-specific effects (e.g., animacy division) observed in these same layers are due to the sensitivity of these layers to represent background information in addition to objects [68–70]. Such questions arise because apparently DCNNs in their final layers, learn representations in which object and scene information is more entangled than what we observe in the human visual cortex. In other words, the final DCNNs layers, represent both the animal and its associated scene (e.g., polar bear and ice landscape), whereas VTC represents animals and scenes in separated areas. One possibility is that DCNNs represent animals separated from inanimate objects because the former is typically seen in specific backgrounds. Taking such shortcuts can be advantageous such as the case

shown here, where DCNNs acquired human-like "knowledge" on which they were not trained. However, in general, studies show that relying too much on background information can result in remarkable non-human-like errors. Examples have been reported where DCNNs accuracy for animal recognition drops in unfamiliar backgrounds [71], large animals go undetected in unlikely environments such as a living room [72], or more simply, object classification might be primarily driven by object-irrelevant information present in the image, which happen to correlate with categorization-relevant features [46].

Previous studies have shown that DCNNs learn a hierarchical representational space that mimics the visual hierarchy observed in the ventral pathway, where the top DCNNs layers are those that best fit with VTC representations [22,73]. Our results go beyond previous observations and show that VTC domain-specific representations are well captured by mid-level layers instead, because final layers appear to even be able to predict high-level representations observed in downstream frontoparietal areas, generally associated to goal-directed behavior [32]. How can we explain these findings? We believe that the answer lies in the selection of the stimulus set. Previous studies have mainly focused on independent category representations (e.g., animate-inanimate), here instead we created a stimulus set that in addition to within-domain relations allows to test similarities across domains that can be found in the co-occurrence of an object in its typical scene. Thus, the crucial difference between our design and that of previous studies is that we manipulated scene congruence, which results in a remarkable dissociation between late DCNN layers (which incorporate this congruence) and VTC regions (which do not).

Although DCNNs might implement different computational strategies they do still develop "rich" internal representations for different object categories [e.g., faces, objects; 74,75,76], and their features [e.g., eyes for faces; 43]. These representations/features are the result of specific tasks DCNNs are trained on (e.g., object recognition). Already in our study, the object space in mid-layers, revealed evidence for a degree of animacy continuum on top of object-scene division. What these representations lack, however, is the diversity and division of labor that we see in the human brain (Fig 6). The representational diversity observed in visual cortex is likely to result from the need of our brain to employ visual information to support diverse behavioral goals over and above object recognition: from recognizing the identity and mental states of people we constantly interact with, to the ability to navigate in the surrounding environment. These computations are implemented in parallel brain networks and the representations in domain-specific networks are optimized to support the different computational goals our brain constantly deals with. Thus, for instance, in visual cortex, scene-selective areas represent scene layout information relevant to support navigation which is not captured by DCNNs trained on scene recognition tasks (Fig 6). DCNNs can be trained in separate domains though. Training a network on a specific object domain (e.g., faces and objects) leads to diverging object spaces in late DCNNs' layers [77], which however, do not generalize well to other domains [74,78,79]. Together, this highlights the advantages of domain-specialized modules evolved by biological vision [51]. We suggest that future DCNNs that aim to capture the rich and diverse representational space found in VTC need to employ tasks that go beyond standard object recognition tasks and target the diverse computational goals our visual system supports [49].

For our DCNNs analyses, we employed a relatively small dataset in comparison with dataset available in computer vision. We acknowledge that this might be a limitation but at the same time, the dataset we tested has the advantage to be carefully controlled for many factors. This is why we can compare multiple competing models. The problem with many datasets of natural images is that many dimensions of interests are naturally correlated. Therefore, in many cases, it becomes difficult to disentangle the relevant dimensions unless creating an ad-hoc stimulus set

which despite its size limitation, allows to tackle specific experimental hypotheses. We therefore believe that to complement results based on big data sets, the need for well controlled (but inevitably small) data sets is of great added value to test cognitive/psychological theories/hypotheses.

In sum, our study confirms and goes beyond previous studies showing a quite remarkable ability of DCNNs to mimic human object vision at the behavioral and neural representational level in a previously unexplored aspect: the representation of object-scene correspondences. At the same time, it demonstrates the importance of a unique aspect of human information processing relative to machine vision: human information processing represents rich and diverse object spaces. Most likely, this is related to the fact that the human brain has evolved and has been trained to implement a wide variety of tasks that require to extract different information type from the different domains present in the visual scene: object information might be more relevant for computations that pertain human-object interactions, whereas information from the background is more relevant to spatial computations such as navigation in the environment. As a consequence, humans understand that an object is not just a collection of features, which could erroneously lead to the assumption that green leaves are inherently linked to ladybugs or to adversarial errors such as classifying a set of yellow and black stripes as a school bus [80]. While the human brain is definitely also prone to the use of shortcuts and heuristics [81], the presence of multiple streams of processing and of a multitude of modular systems has the potential of limiting the impact of such shortcuts. It is a major challenge for the future to develop neural network models with a similar richness of representational content.

## Methods and materials

### Ethics statement

All participants signed the informed consent approved by the ethic committee at the Katholieke Universiteit of Leuven (B322201630276).

### Participants

We recruited 21 participants (7 females, mean age 29). All participants took part in the behavioral test and in 2 functional neuroimaging sessions. Due to technical problems, 1 session from 2 participants and 2 sessions from 1 participant went lost. Additionally, due to excessive head motion, all data from 1 participant and 4 runs from an additional participant were excluded from the data analyses. The head motion exclusion criterion was set to 2 mm (equal to 1 voxel size) and defined before data collection.

### Stimuli

We constructed a stimulus set with contextual-related pairs of images (Fig 1A). For each contextual pair (1 background and 1 animal), we selected 6 conditions, each containing 4 examples. The animal conditions were depicted on a neutral background to avoid low-level associations between animals and their typical backgrounds. To balance the animals' shape information, we included 2 similarly colorful (red) rounded shaped and small size animals (clownfish and ladybug), 2 birds (passerine and seagull) and 2 mammals with similar body shape (polar bear and gorilla). Thus, conditions within each animal pair are matched in terms of taxonomic and visual information (the two birds have a similar shape, visual features, and both fly), but are specifically associated with a different background: one was associated with a watery background (polar bear and ice landscapes) and one with a greenery background (gorilla and jungle). Finally, each stimulus domain is characterized by a behaviorally relevant dimension: the animacy continuum for the animal conditions [fish/bug, birds, mammals; 47]

and the degree of navigational properties [7,8] for the background conditions (seashore, ice landscape, forest jungle versus anemones, leaves, tree branches).

### fMRI data

The fMRI data was acquired in two separate sessions within 2 weeks. Each session comprised 8 runs for the main experiment and 2 runs for the functional localizer. For the main experiment, each run lasted 6 min and 52 sec (206 volumes). For each run, the 48 image trials and 16 fixation trials were presented twice in a random order. Each trial was presented for 1500 ms and followed by a fixation screen for 1500 ms. Each run started and ended with 14 s of fixation. During scanning participants performed a one-back task judging on a scale from 1 to 4 the degree of co-occurrence of each pair of stimuli ("How often do you see these images occurring together?"). The response order was counterbalanced across runs. Before the first scanner session, participants familiarized with the stimuli and performed a similarity judgments task [82] arranging the 48 images (Fig 1A) according to "the degree to which you see these images occurring together". The resulting behavioral dissimilarity matrices were averaged across participants.

**Acquisition parameters.**    The fMRI scans were acquired on a 3T Philips scanner with a 32-channel coil at the Department of Radiology of the University Hospitals Leuven. MRI volumes were collected using echo planar (EPI) T2*-weighted scans. Acquisition parameters were as follows: repetition time (TR) of 2 s, echo time (TE) of 30 ms, flip angle (FA) of 90˚, field of view (FoV) of 224 mm, and matrix size of 112 x 109 and voxel size of 2 x 2 x 2 mm. Each volume comprised 60 axial slices (0.2 mm gap) acquired with a multi-band factor of 2, covering the whole brain. The T1-weighted anatomical images were acquired with an MP-RAGE sequence, with 1 x 1 x 1 mm resolution.

**Preprocessing.**    Before statistical analysis, functional images underwent a standard preprocessing procedure (SPM 12, Welcome Department of Cognitive Neurology) including three-dimensional head-motion corrected (2nd degree spine interpolation), coregistration to the individual anatomical images and normalization to an MNI (Montreal Neurological Institute) template. Spatial smoothing by convolution of a Gaussian kernel of 4 mm full width at half-maximum was applied to functional images [83]. For each participant, a general linear model (GLM) was created to model the 48 conditions and the six motion correction parameters (x, y, z for translation and for rotation). Each predictor's time course was modeled for 3 s (stimulus presentation + fixation) by a boxcar function convolved with the canonical hemodynamic response function in SPM.

**Regions of interest (ROIs).**    All ROIs were defined at the group level based on anatomical masks defined with the Neuromorphometrics SPM toolbox, and functional masks defined with an independent localizer. All visually active voxels (all 48 conditions vs baseline) exceeding the uncorrected threshold p <0.0001 within the functional/anatomical masks were included. The anatomical-based ROIs included 2 visual areas (V1, posterior VTC), and 2 frontoparietal areas, intraparietal sulcus (IPS) and dorsal prefrontal cortex (DPFC). The functionally defined ROIs included animal- and scene-selective areas defined with an independent localizer. Whereas "scene-selective" is standard terminology, "animal-selective" is not. We chose this terminology for the following reason. We know that medial VTC responds more to inanimate objects including scenes, whereas the lateral VTC responds more to animate objects including faces, bodies and animals more in general. For this reason, in a separated localizer, we defined animal- and scene-selective areas based on the direct contrast between these two conditions (animals versus scenes; scenes versus animals). This contrast successfully highlighted animal-selective areas in lateral and ventral VTC and scene-selective areas in parahippocampal gyrus (PPA), retrosplenial cortex (RSC), and transverse occipital sulcus (TOS).

## DCNN data

We extracted the vector spaces for images in our stimulus set from 4 recent DCNNs with different depth: Alexnet [84], VGG-19 [85], GoogLeNet [86] and ResNet 50 [87]. DCNNs consist of various feedforward processing stages including (1) convolutional layers, (2) rectified linear unit activation, (3) max pooling layers, and (4) fully connected layers (resembling a multilayer perceptron). For each DCNNs and each convolutional and fully connected layer, we extracted the vector space for all 48 stimuli by means of the Deep Learning Toolbox (https://uk. mathworks.com/products/deep-learning.html) in MATLAB. All DCNNs were pre-trained in object recognition on 1.2 million natural images belonging to 1000 classes for the ImageNet dataset (http://www.image-net.org/). In addition, GoogLeNet was also pre-trained in scene recognition on the Place365 dataset which contains 10 million images comprising more than 400 scene classes. Each scene contains between 5000 to 30,000 training images.

## Statistical analysis

We used representational similarity analysis [RSA; 29] to compare the neural activity patterns from the brain's ROIs and DCNNs' layers to our experimental models. CoSMoMVPA [88] was used to create the representational dissimilarity matrices (RDMs) for brain and DCNNs data. For neural data, for each voxel within a given ROI, GLM parameter estimates for each stimulus condition (relative to baseline) were extracted for each participant and each run and normalized by subtracting the mean response across all conditions. Similarly, for each DCNNs, the vector features for each stimulus condition within each layer were normalized by subtracting the mean feature activations across all conditions. Next, we computed Pearson's correlation across voxels (fMRI) or units (DCNNs) between all condition pairs and converted the resulting correlation matrix into a representational dissimilarity matrix (RDM; 1 minus Pearson's r) and used it as input for the RSA analysis. We performed two main analyses. To test our main experimental hypothesis, we run two RSAs for both brain and DCNNs data. The first analysis tested two main models: the domain model which assumed high similarity for stimuli within each domain (animals and scenes) and the co-occurrence model which assumed high similarity for each animal-scene contextual-related pair. These two models were derived from our two alternative hypotheses and were orthogonal to each other ($r = -0.05$). Two additional control models were included: the GIST model [27] to capture low-level image information and the condition model to account for within condition similarities. The correlations between all models are the following: (domain and GIST: $r = 0.11$; domain and condition: $r = 0.27$; co-occurrence and GIST: $r = 0.16$; co-occurrence and condition: $r = 0.62$; GIST and condition: $r = 0.23$). For the neural data, the RSA was performed for group averaged data and for single subject data. For the latter one we calculated reliability of RDMs which indicates the highest expected correlation in a brain region given its signal-to-noise ratio. For each subject and each ROI, the RDM was correlated with the averaged RDM of the remaining participants. The resulting correlation values (averaged across participants) capture noise inherent to a single subject as well as noise caused by inter-subject variability. For the second RSA (Fig 6), in addition to the two control models, the following experimental models were included: the animacy continuum model which captures the well-known animacy continuum described in VTC and the navigational layout model which instead captures the degree to which a scene contains features useful for navigation. The animacy continuum model [47] captures the degree to which an animal is perceived as animate (within the animate/inanimate division some animals are perceived more animate than others: mammals > birds, birds > fish/bugs). The navigational layout model captures the degree to which scene layout provides relevant information for scene navigation. 50% of scene images (jungle forests, ice landscapes, and

beaches) contain high degree of layout information. These images are those associated with large animals (e.g., gorilla) and can be considered typical scenes that convey information enough about scene navigation. On the contrary, the remaining images contain little layout information (e.g., leaves, anemoni, and tree brunches). In other words, these images represent zoom-in scenes that from a human point of view provide little information to support navigation. Based on this consideration, the navigational layout model includes a binary distinction between these 2 sets of backgrounds. The backgrounds with low-navigational layout properties are also characterised by object-like properties, as justified by their labels (e.g., 'leaves'), which we also incorporated in the model by a partial similarity between these backgrounds and the animal images. As for the first RSA, two control models were included: (1) the condition model, is considered the default model for the classification tasks employed in computational vision model which predicts classification at the level of basic category (e.g., polar bears, beaches etc), and the GIST model to control for low-level image properties. For all RSAs, we used partial correlation to account for any partial relationship between models. Before statistical analyses, results from the RSAs were Fisher transformed ($0.5*\log[(1 + r)/(1—r)]$) and tested with ANOVAs and pairwise t tests when individual-subject data were available, or permutation tests for DCNNs and fMRI group averaged data.

In addition to ROI-based RSAs we performed two whole-brain RSAs. The whole-brain RSA was implemented in CoSMo MVPA (Oosterhof et al., 2016) using the volume-based searching approach (Kriegeskorte et al., 2006). For each condition (relative to baseline), parameter estimates were extracted for each participant and each run and normalized by subtracting the mean response across all conditions. Resulting values were then averaged across all runs. For each brain voxel, a searchlight was defined using a spherical neighborhood with a variable radius, including the 100 voxels nearest to the center voxel. For each searchlight, the neural dissimilarity matrix was computed for the 48 stimuli. The neural dissimilarity matrix (upper triangle) was then correlated with the dissimilarity matrices derived from the 4 models (Fig 1C) and dissimilarity matrices derived from all layers of AlexNet. The output correlation values were Fisher transformed and assigned to the center voxel of the sphere. Resulting whole-brain correlation maps for each model and layer were tested against baseline using random effects whole-brain group analysis corrected with the threshold-free cluster enhancement (TFCE) method [37]. Voxelwise-corrected statistical maps for each model/layer relative to baseline (z 1.96; $p < 0.025$, one-sided t test) are displayed on a brain template by means of BrainNet Viewer [38].

## DCNN experiment with congruency manipulation

Construction of the training datasets: Custom datasets were constructed by joining masked images of objects with inverse-masked images of scenes obtained from two publicly available datasets. Images of 6 object categories ('bus', 'clock', 'dog', 'giraffe', 'television', 'umbrella') were obtained from the Microsoft COCO dataset that consists of over 300.000 images labelled with more than 2 million object instances [89]. Each object instance is associated with a mask which allows extraction of object-representing pixels. Objects of small size (ratio between object size versus total image size smaller than 3%) were enlarged and objects of object-image ratio smaller than 0.15% omitted in order to avoid unsatisfactory classification performance on images with hardly any object-associated pixels [90]. Scene images of 6 categories ('church indoor', 'sand desert, 'glacier', 'ocean underwater', 'broadleaf forest', 'landfill') were obtained from the Places dataset that consists of more than 2 million images of 205 scene categories, most of them containing several thousand unique images [91]. This allowed generation of a total of 10500 training images (1750 per object category) of 256 x 256 x 3 pixels per dataset, in which each image showed an object of one of the 6 object categories embedded in a scene of

one of the 6 scene categories. Note that with only 6 image classes with a total of 10500 training images we are not claiming that the resulting DCNN models will be usable as state-of-the-art models in image classification or as reference models to compare with neural representations. Instead, these models are merely developed to provide a proof-of-principle showing how strong the impact of scene-object congruency in a training set that is controlled in terms of this congruency, in contrast to typical training sets such as ImageNet.

Four different datasets of 0%, 58.3%, 83.3% and 100% of object-scene co-occurrence were constructed by manipulating the likelihood of objects being joined with their adhering background type. Each dataset was constructed out of the same set of object and background images that were, however, joined in a different fashion. To illustrate this on the example of 83.3% object-scene co-occurrence dataset, backgrounds of every 1st, 2nd, 3rd and 4th image were object-specific (80%) and background of every 5th image was belonging to either one of the six background categories (20%). Thus, there is an additional 1/6 chance that randomly chosen backgrounds will come from the object-specific category (adding additional 3.3% to the total object-scene specificity of the dataset). In the 0% object-scene co-occurrence, none of the 6 scene categories is more likely to be associated with a particular object category than the other scene categories (all have a 16.7% chance).

**Training of the models.**   Five individual DCNN models were trained on each of the constructed datasets. The models followed the standard AlexNet architecture with five convolutional and three fully-connected layers with one minor difference in input layer accepting images of size 256 x 256 x 3 instead of 224 x 224 x 3 pixels, as originally defined by Krizhevsky and colleagues (2012)[84] to better fit the size of our image stimuli. All models were trained for 150 epochs in batches of 75 training images, with the Adam optimizer and categorical cross-entropy loss function. A dropout regularization rate of 0.5 was added to the first two fully-connected layers. Additionally, image augmentation was employed to improve generalization of the trained models. At the end of the training, only the weights of the best performing epoch (lowest validation loss calculated on a total 750 validation images, 125 per object category) were stored. All models were implemented in Python utilizing functions of the Keras library.

**Representational similarity analysis with congruency manipulation.**   Additionally, representational similarity analysis was performed to compare patterns of inner neural representations of the trained networks with our experimental models. In order to do so, neural activations of each networks' 7th layer (fully connected) were extracted by presenting them with 48 image stimuli, each sprouting 4096 neural activations. This set of stimuli consisted of 24 images of objects on neutralized backgrounds and 24 images of scenes without the object entities (4 for each object category and 4 for each scene category). Neutralized backgrounds were obtained by averaging pixels of 6 scene images randomly chosen from Places dataset [91], each of them belonging to one of the above specified scene categories. This was done in order to provide objects with a background that possessed no relevant scene information yet were not too atypical/homogeneous and thus too different from training examples (usage of monotonous backgrounds was found to be problematic as it significantly impacted internal representations of the networks). Next, correlations between internal activations for each stimulus were obtained by calculation of Pearson correlation coefficients. This resulted in a single similarity matrix for each model which was in the end transformed into an RDM matrix by subtracting it from a unit matrix (with all values equal one) of a matching dimension.

## Supporting information

**S1 Fig. Dissimilarity matrices computed with alternative distance metrics.** The main brain RSA analysis is replicated with alternative distance measures: (A) cross-validated Mahalanobis

distance following [1] and (B) Euclidean distance. As for the main RSA, we tested 4 models: GIST, condition, domain, co-occurrence. Results confirm data analysis performed with 1-corr distance. Filled bars indicate significant values against baseline (p <0.005, corrected for n. or ROIs) calculated with pairwise t-tests across subjects (n = 19).
(DOCX)

**S2 Fig. VTC representations are best captured by mid-level DCNNs layers.** (A) For each DCNNs, the RSA results show the degree of similarity between the representational space of individual layers in each brain area. (B) The random-effects whole-brain RSA results corrected with Threshold-Free Cluster Enhancement [TFCE; 2] are displayed separately for each layer of AlexNet against baseline [BrainNet Viewer; 3].
(DOCX)

**S3 Fig. Domain-specific spaces tested within-domain data.** The RSA results for the 4 models (GIST, condition, animacy continuum, navigational layout) are shown for group-averaged brain (left) data and DCNNs (right). The same DCNN architecture (GoogLeNet) was trained either on object recognition (ImageNet), or scene recognition (Scene 365). For comparison with the first RSA analysis (see Fig 3A), gray shaded areas indicate the network's layers in which the domain model significantly outperformed the remaining models. Color-coded lines on top of bar/graphs indicate the network's layers/ROIs where each model significantly out-performed the remaining models (p < 0.001) computed with pairwise permutations tests (10000 randomizations of stimulus labels).
(DOCX)

## Author Contributions

**Conceptualization:** Stefania Bracci, Hans Op de Beeck.

**Data curation:** Stefania Bracci, Astrid Zeman, Gaëlle Leys.

**Formal analysis:** Stefania Bracci, Jakob Mraz.

**Funding acquisition:** Stefania Bracci, Hans Op de Beeck.

**Investigation:** Stefania Bracci.

**Methodology:** Stefania Bracci.

**Project administration:** Stefania Bracci.

**Supervision:** Hans Op de Beeck.

**Visualization:** Stefania Bracci.

**Writing – original draft:** Stefania Bracci, Jakob Mraz, Hans Op de Beeck.

**Writing – review & editing:** Stefania Bracci, Jakob Mraz, Hans Op de Beeck.

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
