## [Decision Letter · Decision Letter 0]

28 Jul 2022

Dear Dr. Bracci,

Thank you very much for submitting your manuscript "The representational hierarchy in human and artificial visual systems in the presence of object-scene regularities" for consideration at PLOS Computational Biology.

As with all papers reviewed by the journal, your manuscript was reviewed by members of the editorial board and by several independent reviewers. In light of the reviews (below this email), we would like to invite the resubmission of a significantly-revised version that takes into account the reviewers' comments.

We cannot make any decision about publication until we have seen the revised manuscript and your response to the reviewers' comments. Your revised manuscript is also likely to be sent to reviewers for further evaluation.

Sincerely,

Tianming Yang

Associate Editor

PLOS Computational Biology

Thomas Serre

Deputy Editor

PLOS Computational Biology

Reviewer's Responses to Questions

**Comments to the Authors:**

Reviewer #1: This paper investigates the separate and conjunctive representations of scenes and objects in the brain and deep neural network models trained for visual object recognition. A set of four models are used to capture different hypotheses on the nature of the representation. By comparing the brain activity in different regions of the ventral temporal and parieto-frontal regions, it is shown that how each area represents objects (animate) and scenes in various ways. The rest of the paper then takes a deep dive into comparing different DNN architectures on the degree of brain-similarity in representing these two category of stimuli. In summary, similar to the brain, different DNNs also show similar representations for coding objects and scenes and their interactions, capture human-like representational biases in coding scenes and objects, but fail to replicate some aspects of the brain representation - namely "multiple object spaces".

Overall, this was an interesting and well written paper that was easy to follow. The stimulus set and the accompanying fMRI experiment were designed thoughtfully. The modeling experiments were also justified and were used to support specific claims. I however had some concern with the particular method used for investigating representational similarity and found parts of the manuscript unclear.

- The present study uses a correlation-based RSA metric to investigate similarity between different models and the brain. It is known that correlation distance metrics are sensitive to stimulus activations, unlike Euclidean distance and its derivatives [1]. It is therefore unclear to me whether the chosen metric is the best metric for the analyses carried out throughout the manuscript and to what degree the findings are dependent on this choice of metric. Can the authors show that for example by choosing a Euclidean distance metric, the claims and findings of the paper remain valid?

- It is unclear why the subjects were instructed to "indicate to what extent each image would normally co-occur with the previous image". Could this task not bias the brain representations in regions underlying decision making like PFC to be biased towards interactive effects like those observed in the study? Why hasn't a simpler task like passive fixation used instead? The logic here was unclear to me.

- It is unclear what "animal-selective areas" was referring to. Are all VT regions considered animal-selective?

- In figure 3, the ordering of brain ROIs in Fig. 3B implicitly suggests a particular hierarchy between them and lack thereof in the models. However the hierarchy of these brain areas are not as strictly linear and thus this part of the figure may be misleading.

- In Figure 4, resnet50 model does not seem to follow one of the claims associated with this figure: "representational correspondence with frontoparietal areas increases and remains high till the final processing. This effect was observed regardless of DCNNs architecture.". This divergence should be discussed in text.

- The notion of "multiple object spaces" was somewhat unclear to me, especially in the context of DNNs. It is known that these models contain category-specific kernels that are selective for particular objects or object parts. The last section of the results seems to claim that DNNs lack "rich domain-specific object spaces". I found this claim to be contradicting more than several other works highlighting the richness of these representations e.g. see [2, 3]. Perhaps what is meant to be claimed is specific to the representational of navigational layout? If yes, this point needs to be better clarified. Even so, can this failure to find similar detailed distinction across the scene stimuli be due to the high dimensionality of these networks and the limitations of the method (RSA) used?

- Related to the previous point, it was unclear how the "navigational layout" in Fig7A produced. There are 4 columns that presumably correspond to 4 of the scene stimuli (?).

- Line 520: it is claimed that "object and scene information is more entangled than what we observe in the human visual system". The evidence supporting this statement is unclear to me. Please add appropriate references and discussion.

- Also line 566-568: it is claimed that machine vision lacks "rich and diverse object spaces". It is unclear in what ways this feature is missing in current DNNs

- I'm also somewhat concerned with the sufficiency of the dataset used to train the DNN models for Fig 6. In the methods section, it is mentioned that a total of 10,500 stimuli were used to train the DNN. Considering that there are around 60 million parameters in this model, the dataset of this size sounds way too small and I'm worried about the findings from this experiment. It would be helpful to see a plot of training and validation losses for these experiments as well as examples of some of these images used for training/validation.

- typos: line 230, what -> which; line 287: in -> and; 412: an -> a

1. Walther, A. et al. Neuroimage 137, 188–200 (2016).

2. Cohen, U., Chung, S.Y., Lee, D.D. & Sompolinsky, H. Nat. Commun. 11, 1–13 (2020).

3. Murty, N. A. R., Bashivan, P., Abate, A., DiCarlo, J, and Kanwisher, N. Nat. Commun. (2021).doi:10.1038/s41467-021-25409-6

Reviewer #2: In Bracci et al’s paper, the authors investigated the representation of the animals and their living environmental images in the human brain with fMRI, as well as the DCNN. They not only found the areas that represent the within-domain, but also the ones representing the across-domain, which indicates the co-occurrence of the animals and their living environment. The Similar representations of the within-domain and across-domain were also observed in the DCNN and the latter was dependent on the training image-sets. In addition, they also tested the animal continuum model and navigation layout model in human brain and DCNN. They confirmed that the animal continuum model was represented in both, while the navigation layout model only existed in the human brain.

In general, I think the whole experiment is well designed and some of the results are very interesting. The whole paper is well written and clear. However, I still have some comments/questions:

1) I am not sure I understand navigation spatial layout RDM model in Fig. 7A. For the animal continuum model, I understand that the authors consider the natural scene images as the non-animal images. But for the navigation spatial layout RDM, they consider the first two (or three?) groups of scene images are similar to animal images, but have zero similarity with the other scene images. How do the authors determine such RDM structure? Also for the animal continuum model, it is highly similar to the within-domain model structure. So, given the previous results, it is not surprising that they can find the animal continuum representation in the brain and DCNN. They might consider testing the animal continuum model only with animal images. The same analysis could be also applied with the spatial layout model. In addition, I think the whole analysis is Figure 7 is not that convincing, due to the small number of images and test conditions. For the animal continuum model, they only have 3 different conditions within the animal images, and two conditions within the scene images. They can test with more images with more conditions at least with DCNN.

2) In Figure 2, it is very interesting to see the effect of animal-scene co-occurrence emerges in frontoparietal areas. But given the low correlation value, I am not sure whether it really reflects co-occurrence. One way to test it is to build the wrong co-occurrence models (e.g., gorilla associated with landscape, and polar bear associated with forest), and then tested the significance level between the correct co-occurrence model and wrong co-occurrence models. Such analysis can be applied with ROI analysis and search light analysis to further confirm that the frontoparietal areas are really representing the co-occurrence structure that exists in the natural world.

Besides these two main points, I have some other minor points:

1) In Keyword part, Line 462,659, DCCN should be DCNN.

2) In line 703, the equation for fisher transformation is wrong.

3) In Fig.2AB, 3A 6B, 7, I don’t understand why the name for Y axis is “partial correlation”. And there is no explanation for “partial correlation” in the method part.

Reviewer #3: The article consists of 3 separate data-sets, each related to the question how animal and scene perception interact. A BOLD-MRI part, a DNN part with the same stimuli, and a DNN part with an extended data-set.

The topic is timely, and the article ambitious with its scope. Papers that manipulate images to test DNNs are only starting to appear and are likely one of the keys to connect psychology with AI. In particular, the results depicted in figure 3, figure 5 and figure 6 are gorgeous! Congratulations, nice work indeed.

However, I have some worries about the design of the MRI experiment, size of the stimulus set and the overall approach used for testing RDMs.

Specifically:

* what is the correlation between the condition and co-occurance model? This is less relevant for the partial correlation, but comes into play in the whole brain results of figure 2. Does co-occurence significantly differ from condition? This should either be an explicit test, or the presentation should be removed (otherwise suggestive of something that is not significant).

* The stimuli used in part 1 and 2 of the paper are limited in size. The conclusions that the authors want to draw on the basis of fig 3 are substantial. It should be possible to do an internal replication of these results with a additional set of stimuli. Preferable of different animals and background scene's.

* In general the RDM comparison approach the authors use is somewhat dated. Would it not be better to use a cross validation approach as used by for instance Storrs et al., 2020 (Diverse deep neural networks all predict human IT well, after training and fitting), or at least crossvalidation in general? Alternatively the authors should argue why not. This is the current state of the art and with good reason.

* The whole brain analysis in figure 4b are correlations, and not differences between layers, however, the results are discussed in those terms (likely the differences will turn out to be not significant).

* The second data-set is more extensive than the first data-set, however the results presented in figure 7 on navigational lay-out are likely still underpowered.

Minor comments:

* I miss the noise ceiling for BA17 and RSC for figure 2

* There is a smooth overlap between scenes with lots of objects and a typical background. The scenes the authors use are somewhat more than a typical textures background but certainly not complex scenes. It would be good if the authors comments on this.

* What is the added value of figure 2a above figure 2b (and 2b has proper statistics).

* Yes, DPFC dips less (fig 4) at the end than OPA, but OPA is still the better model for the top layer, So what point do the authors want to make? Also, are these differences significant?

**Have the authors made all data and (if applicable) computational code underlying the findings in their manuscript fully available?**

Reviewer #1: **No: **the stimulus set and statistics data will be made available through OSF. The plan for sharing the code is not discussed.

Reviewer #2: Yes

Reviewer #3: **No: **The war data is not immediately open because of ethical reasons. However, the data is on OSF, and the authors indicate that on request researchers will be added to the full data. So this should be OK.

PLOS authors have the option to publish the peer review history of their article (what does this mean?). If published, this will include your full peer review and any attached files.

Reviewer #1: **Yes: **Pouya Bashivan

Reviewer #2: No

Reviewer #3: **Yes: **H Steven Scholte
---

## [Decision Letter · Decision Letter 1]

1 Mar 2023

Dear Dr. Bracci,

Thank you very much for submitting your manuscript "The representational hierarchy in human and artificial visual systems in the presence of object-scene regularities" for consideration at PLOS Computational Biology. As with all papers reviewed by the journal, your manuscript was reviewed by members of the editorial board and by several independent reviewers. The reviewers appreciated the attention to an important topic. Based on the reviews, we are likely to accept this manuscript for publication, providing that you modify the manuscript according to the review recommendations.

Sincerely,

Tianming Yang

Academic Editor

PLOS Computational Biology

Thomas Serre

Section Editor

PLOS Computational Biology

Reviewer's Responses to Questions

**Comments to the Authors:**

Reviewer #2: I have no other further questions.

Reviewer #3: The authors have gone over the comments of myself and the other reviewers with diligence. Most of my comments were fully addressed.

For point 1: I understand that the authors would like to keep fig 2c and they have now added additional text in the paper. Would it not be fair to add: the conditions do not differ from each other in the caption of figure 2c?

Nice paper!

For crossing the t's, the disclosure of the data and code is not entirely clear. The MRI data can be requested from the authors? It is not clear if the analysis code will be made available (currently not). Can this be rectified?

**Have the authors made all data and (if applicable) computational code underlying the findings in their manuscript fully available?**

Reviewer #2: Yes

Reviewer #3: **No: **The authors indicate that the stimulus and statistics data will be made available through the Open Science Framework. This is not the case yet (the paper has been created but it is still empty as far as I can see). Also the authors do not indicate that they will share the code.

PLOS authors have the option to publish the peer review history of their article (what does this mean?). If published, this will include your full peer review and any attached files.

Reviewer #2: No

Reviewer #3: **Yes: **H.Steven Scholte

Figure Files:

Data Requirements:

Reproducibility:

References:

---

## [Editor Report · Decision Letter 2]

9 Apr 2023

Dear Dr. Bracci,

We are pleased to inform you that your manuscript 'The representational hierarchy in human and artificial visual systems in the presence of object-scene regularities' has been provisionally accepted for publication in PLOS Computational Biology.

Best regards,

Tianming Yang

Academic Editor

PLOS Computational Biology

Thomas Serre

Section Editor

PLOS Computational Biology

---

## [Editor Report · Acceptance letter]

25 Apr 2023

PCOMPBIOL-D-22-00906R2 

The representational hierarchy in human and artificial visual systems in the presence of object-scene regularities

Dear Dr Bracci,

I am pleased to inform you that your manuscript has been formally accepted for publication in PLOS Computational Biology. Your manuscript is now with our production department and you will be notified of the publication date in due course.

With kind regards,

Timea Kemeri-Szekernyes
